# Is Motivation Associated with Mental Fatigue during Padel Trainings? A Pilot Study

**Jesús Díaz-García ***, **Miguel Ángel López-Gajardo**, **José Carlos Ponce-Bordón** and **Juan José Pulido**

Faculty of Sport Sciences, University of Extremadura, 10003 Cáceres, Spain; mianlopezg@gmail.com (M.Á.L.-G.); jponcebo@alumnos.unex.es (J.C.P.-B.); jjpulido@unex.es (J.J.P.)
* Correspondence: jdiaz@unex.es; Tel.: +34-689-914-911

**Abstract:** Motivation seems to enhance athletes' mental efforts, but this has not been tested yet in padel. The objective was to test the effects of motivation on mental fatigue during padel trainings. Thirty-six elite youth players participated (twenty-two males, $M_{age}$ = 17.40, $SD_{age}$ = 2.16, and fourteen females, $M_{age}$ = 17.90, $SD_{age}$ = 3.21). We designed four padel training matches, introducing a constraint in two of them in a counterbalanced order. The constraint was: Couples that win more sets in these two matches obtain a free lesson with a professional padel player. Motivation was quantified by a questionnaire before the matches. Moreover, subjective feelings of mental load and fatigue were measured with questionnaires, and objective measures of fatigue were quantified through heart-rate variability and reaction time. Results suggest that the constraint significantly increases motivation ($p < 0.001$). Furthermore, in these matches, players reported significantly higher feelings and objective measures of fatigue ($p < 0.001$ for HRV and VAS; $p = 0.04$ for reaction time). An increase in the resources used by the neural facilitation system, mediated by higher values of motivation, seems a relevant candidate to explain this phenomenon.

**Keywords:** HRV; reaction time; VAS; ecological constraints

## 1. Introduction

The evolution of padel has triggered scientist to start studying this sport. It has been shown that padel matches imply both physical and mental demands [1,2]. This suggests that during padel trainings, a synergizing of mental and physical aspects must appear [1]. However, to our knowledge, no previous studies have quantified the mental demands of padel trainings or the influence of different constraints on these demands. This information might help coaches to understand how to use ecological training strategies to modify mental efforts during the trainings according to their objectives [3]. Previous studies have suggested that an increase in motivation could enhance athletes' mental efforts [4,5], although more studies are necessary to test this hypothesis. The main purpose of this study was to test the effects of motivation on mental efforts and fatigue during padel trainings.

The physical demands in padel have been well defined in several studies. They are caused by intermittent efforts that include turns, jumps, accelerations-decelerations, and strokes [1,2,6–8]. However, few works have studied the mental demands caused by padel training or matches. Mental fatigue in sports can be produced both by cognitive and emotional demands [9]. Cognitive demands in padel have been associated with tactical decisions and attention, whereas emotional demands could appear due to match situations (e.g., anxiety due to consecutive fails) or they could be caused by the interdependence of teammates [10]. Although a negative influence of mental fatigue on padel performance has not yet been tested, previous studies performed in other racket sports have reported impairments in technical accuracy [11] and visuomotor responses [12] due to induced acute mental fatigue.

Thus, this information suggests the importance of controlling and manipulating mental fatigue during padel trainings to improve players' tolerance during competitions. In several

sports, which include padel, there has been an increase in ecological training [13]. Applied to padel, this has implied that the mental and physical aspects should be mainly trained in specific padel situations. Using constraints, padel coaches could change the demands of the training tasks, but more studies are necessary to know the true effects of each constraint on mental load [3]. Motivation has probably been the most studied variable (or constraint) in this topic. Herlambang et al. (2019) suggested that high levels of motivation could mitigate the negative effects that mental fatigue has on performance through an increase in the perception of tolerance for effort [14]. Previous studies [5,15] show that high levels of motivation increase the neural facilitation system, enhancing mental efforts and mental fatigue tolerance. However, it also meant higher perceptions of mental fatigue at the end of the training session. Specifically, in the self-determination theory, where athletes' actions can be explained by their interests or external punishments or rewards, it has been shown that constraints can be used to manipulate both extrinsic and intrinsic motivation [14].

On the basis of this information, we suggest that manipulating motivation through constraints might influence the mental efforts of padel players during trainings. The purpose of this study was to test the effects of increased motivation on mental efforts and fatigue during padel training matches. A secondary objective of this study was to confirm the validity of including a free lesson with a professional padel player to manipulate motivation in youth padel players.

Specifically, we hypothesized that high levels of intrinsic motivation would increase mental load and fatigue after a padel training, based on the information provided by Herlambang et al. (2019), Ishii et al. (2014), and McMorris (2020).

## 2. Materials and Methods

### 2.1. Sample

A priori sample size calculation using G*Power was performed. This analysis indicated a minimum sample of 30 padel players, based on the information provided by Batterham and Hopkins (2006) about the effect size (0.20) and the statistical analysis associated with our procedures [16]. Thirty-six elite youth players participated in the study (twenty-two males, $M_{age}$ = 17.40, $SD_{age}$ = 2.16, and fourteen females, $M_{age}$ = 17.90, $SD_{age}$ = 3.21). All of them participate in the youth Spanish Padel Federation circuit (TyC) and have trained at least four days per week for the last three years.

### 2.2. Instruments

Situational Motivation Scale (SIMS). Motivation toward padel trainings was assessed using the Spanish version of the Situational Motivation Scale (SIMS) [17]. This Spanish version includes a total of 14 items: Intrinsic motivation (four items), identified regulation (three items), external regulation (three items), and amotivation (four items). Each question referred to the specific padel training (i.e., why was I participating in this padel training?). Participants rated their responses from 1 (does not correspond at all) to 7 (totally corresponds).

Polar RS800CX. To quantify heart-rate variability (HRV), four reliable heart-rate monitors (Polar RS800CX, Kempele, Finland) were used [18]. Similar to Fuentes et al. (2018), we included the variables heart-rate mean RR mean (HR), standard deviation of all NN intervals (SDNN), NN50 count divided by the total number of all NN intervals (Pnn50), and the square root of the mean of the sum of the squares of the differences between adjacent NN intervals (rMMSD) [19]. All these variables were obtained immediately after each match had finished.

Questionnaire to Quantify Mental Load (QQML). To quantify the mental load, a validated padel-adapted version of the Questionnaire to Quantify the Mental Load in sports was used [20]. It was composed of four items: Perceived physical effort (How much physical activity was required?), cognitive demands (How much cognitive effort was required?), difficulty to control emotions (How difficult is it to control your emotions?), and pressure caused by the interdependence level (How much affective effort was required?).

The responses were rated on a Likert scale ranging from 0 (no perceived effort) to 10 (maximum possible perceived effort).

Visual Analogue Scale (VAS). The 10-cm Visual Analogue Scale was used to quantify the mental fatigue reported by players. Padel players were asked to indicate the perceived level of mental fatigue by marking the VAS 10-cm line. The left side indicated *"not at all"*, while the right side indicated *"maximum"*. This scale has been used in previous studies that quantify mental fatigue reported in sport contexts [11].

Psychomotor Vigilance Task (PVT). The 3-min version of the Psychomotor Vigilance Task was used to measure the reaction time [21]. When a visual stimulus appears in the center of the screen, players have to press the button. Reaction time was calculated as the difference between the appearance of the stimulus and pressing the button.

### 2.3. Procedures

All participants were informed about the objectives of the study, and their parents (under 18 players) signed a signed participation consent, according to the local Ethics Committee (protocol number: 93/2020).

Researchers contacted the training club of these players. Their coaches indicated that these players played training matches each Friday (their final training of the week). We designed a quasi-experimental pilot study, including four sessions where the constraint was introduced in a counterbalanced order. The four matches were played by the same players to avoid the effect of playing against different opponents. The constraint designed consisted of: Players who win more sets after the two constraint-matches win a free lesson with a World Padel Tour player. The players were informed about this constraint before the start of these two matches only, whereas no constraints were used in the results of the other two unconstrained matches. Before the start of each match, SIMS was measured, and QQML, VAS, and PVT were measured after each match.

The official padel rules described by the Spanish Padel Federation were applied during the matches. However, to avoid the effect of match-duration, in all matches, three sets were played (in normal conditions, when a couple wins two sets, the match is over). If we had designed normal conditions, our results could be influenced by the differences between two- and three-set matches. Moreover, to maintain the intensity (in the case that a couple won 2-0), we indicated that the free lesson with the World Padel Tour player would be obtained by the couple who won more sets after the global results of these two constraint-matches.

### 2.4. Statistical Analysis

Statistical analyses were performed with SPSS 25.0. Data were expressed as mean ± SD both of constraint-matches and unconstrained matches (e.g., we expressed the mean values of the two constraint-matches). The Shapiro–Wilk normality test was performed. Sphericity was verified by Mauchly's test. When sphericity was not met, *F* ratios' significance was adjusted with the Greenhouse–Geisser procedure. Levene's test was used to evaluate the equality of the variances. One-way MANOVA was performed to determine the difference in all these variables between constraint and unconstrained training matches, using the presence or absence of the constraint as a factor. For further analysis, Hopkins' spreadsheet was also used. The magnitude of change, considered the effect size (ES), was also calculated [22]. Following Batterham and Hopkins (2006), ES were classified as: *trivial* (<0.2), *small* (0.2–0.6), *moderate* (0.6–1.2), *large* (1.2–2.0), and *very large* (>2.0). Magnitude-based inferences (MBI), with confidence intervals, determined the change in the dependent variables. The smallest worthwhile change (SWC) was set at ES = 0.2 [16]. A qualitative analysis of the change using Batterham and Hopkins' (2006) classification was also performed: 0.5 to 5%, *very unlikely*; 5 to 25%, *unlikely*; 25 to 75%, *possibly*; 75 to 95%, *likely*; 95 to 99.5%, *very likely*; and >99.5%, *most likely*.

## 3. Results

Table 1 shows the results of the motivation-reported scores before playing the padel matches designed. These results reveal that both intrinsic and extrinsic motivation were significantly higher before the constraint-matches in comparison with the unconstrained matches. However, identified regulation did not show significant differences between the two conditions, whereas significantly lower values of amotivation were observed in the constrained matches than in unconstrained matches. These results suggest that the use of this constraint could increase both intrinsic and extrinsic motivation before the start of padel training matches.

**Table 1.** Analysis of the situational motivation and the influence of the constraint.

| Variables | Unconstrained | Constrained Matches | MANOVA/Effect Size Analysis |
|---|---|---|---|
| Intrinsic motivation | $5.17 \pm 1.47$ | $6.31 \pm 1.98$ | F = 12.17, $p$ = <0.001 ***/ES = 0.65; QI = Most Likely + ive (100/0/0) |
| Identified regulation | $4.36 \pm 1.13$ | $4.55 \pm 1.16$ | F = 1.44, $p$ = 0.37/ES = 0.16; QI = Unclear (90/0/10) |
| External motivation | $3.86 \pm 1.17$ | $4.45 \pm 1.29$ | F = 10.71, $p$ = <0.001 ***/ES = 0.48; QI = Most Likely + ive (100/0/0) |
| Amotivation | $2.67 \pm 0.98$ | $2.21 \pm 0.94$ | F = 8.86, $p$ = <0.001 ***/ES = $-0.47$; QI = Most Likely $-$ ive (0/0/100) |

*Note.* *** $p$ < 0.001. ES = Effect Size. QI = Qualitative Inference.

Table 2 shows the results of the subjective and objective measures of fatigue caused by the padel training matches. These results show that padel players reported both significantly higher scores of mental load and subjective feelings of mental fatigue after the constrained matches than in unconstrained matches. The objective measures of central fatigue as Mean HR, SDNN, Pnn50, and rMSSD showed significantly higher values after constrained matches than in unconstrained matches. Moreover, significantly poor values of reaction time were registered after constrained matches. These results suggest that the use of this constraint could increase both objective and subjective indicators of fatigue, also accompanied by longer reaction times.

**Table 2.** Analysis of mental effort and fatigue and the influence of the constraint.

| Variables | Unconstrained | Constrained Matches | MANOVA/Effect Size Analysis |
|---|---|---|---|
| Mental load | $6.45 \pm 1.49$ | $8.67 \pm 2.27$ | F = 20.35, $p$ = <0.001 ***/ES = 1.15; QI = Most Likely + ive (100/0/0) |
| Mental fatigue | $5.67 \pm 1.28$ | $7.08 \pm 2.29$ | F = 14.47, $p$ = <0.001 ***/ES = 0.76; Most Likely + ive (100/0/0) |
| Reaction time | $0.43 \pm 0.02$ | $0.49 \pm 0.02$ | F = 5.72, $p$ = 0.04 */ES = 0.51; QI = Most Likely + ive (100/0/0) |
| Mean HR | $137.67 \pm 14.89$ | $146.19 \pm 16.43$ | F = 6.94, $p$ = 0.02 */ES = 0.54; QI = Most Likely + ive (100/0/0) |
| SDNN | $38.76 \pm 8.82$ | $31.21 \pm 6.91$ | F = 15.86, $p$ = <0.001 ***/ES = $-0.81$; QI = Most Likely $-$ ive (0/0/100) |
| Pnn50 (%) | $10.22 \pm 1.78$ | $6.52 \pm 1.23$ | F = 23.65, $p$ = <0.001 ***/ES = $-2.41$; QI = Most Likely $-$ ive (0/0/100) |
| rMSSD | $22.65 \pm 6.98$ | $19.48 \pm 5.91$ | F = 5.43, $p$ = 0.04 */ES = $-0.49$; QI = Most Likely $-$ ive (0/0/100) |

*Note.* * $p$ < 0.05. *** $p$ < 0.001. Mean HR = Mean heart rate; SDNN = Standard deviation of all NN intervals; Pnn50 = NN50 divided by the total number of NN intervals; rMSSD = square root of the mean of the sum of the squares of the differences between adjacent NN intervals ES = Effect size. QI= Qualitative Inference.

## 4. Discussion

The main purpose of this study was to test the effects of motivation increases on players' mental efforts in padel training matches. Overall, results showed that obtaining a free lesson with a World Padel Tour player by winning padel training matches could be a valid constraint to increase both intrinsic and extrinsic motivation before the start of these padel training matches. Moreover, due to the introduction of this constraint, players reported higher subjective perceptions of mental load and fatigue. Furthermore, HRV measures showed higher objective measures of fatigue when using these constraints.

Therefore, our results were in line with the hypothesis, in which we stated that motivation increases would produce an increase both in subjective and objective measures of fatigue.

Concerning the analysis of the situational motivation and the influence of the constraint, padel players reported higher values of intrinsic and extrinsic motivation before constraint-matches. It suggests that obtaining a free lesson with a World Padel Tour player is potentially motivating for youth players. Certainly, the constraint used could be classified as an extrinsic reward because it was obtained by winning a designed task. Previous studies performed in sports have revealed that external benefits (e.g., monetary rewards) increase athletes' motivation [14,23]. However, these results suggest that receiving a padel lesson from a professional player was also intrinsically motivating for the players. Increases in motivation associated with intrinsic benefits have also been reported [24]. These results have an important practical application considering the effects of this constraint on motivation, and it could be applied by padel coaches when their training objective is to increase both the intrinsic and extrinsic motivation of their youth players.

With regard to the analysis of mental effort and fatigue and the influence of the constraint, the higher values of motivation produced by this constraint were accompanied by an increase both in the objective and subjective indicators of mental load and fatigue. This suggests that the function of the neural facilitation system improved in the presence of this constraint [5,15]. These results are in line with the conclusions of Herlambang et al. (2019), supporting the idea that tasks with higher values of motivation imply higher efforts by players because they involve more resources to solve the task. Ishii et al. (2014) explained the neural mechanism that supports this phenomenon. These authors reported that an increase in the resources of the facilitation system (e.g., catecholamines) could be activated by high levels of motivation to maintain the performance (also in the presence of fatigue). Athletes are assumed to finish their sports activities with higher levels of fatigue, due to an overload of this facilitation system. Therefore, the increase in the efforts produced by the facilitation systems seems to explain why high levels of motivation could increase players' mental effort and fatigue during padel training matches.

To summarize, padel players may attach more importance to solving a task that implies higher levels of motivation for them. These tasks involve more neural resources, and this leads to higher values of fatigue as a consequence. We can only suggest this conclusion, due to the few training sessions and the homogeneity of the subjects, which are the main limitations of the study. Although this should be tested in other contexts (e.g., professional players) for a high number of training sessions, important practical applications could be derived from these results. It has been shown that padel players need to train both physical and mental aspects. On the one hand, there have been important advances both in real and simulated contexts [25]. On the other hand, it has been shown that motivation may be a determinant during these trainings. The present study showed how increasing motivation could enhance players' mental efforts. Therefore, motivation (among other constraints) could be manipulated by padel coaches during their training sessions treat to increase their players' mental efforts.

## 5. Conclusions

To conclude, this is the first study testing an increase in motivation to determine its effects on mental efforts during padel training matches. Subjective feelings of mental load and fatigue measured with QQML and VAS were significantly enhanced in the presence of higher values of motivation. HRV values were also significantly higher when motivation increased. In addition, receiving a free lesson from a World Padel Tour player seems to increase intrinsic and extrinsic motivation. These results add to our knowledge about how padel coaches can manipulate mental load and fatigue in ecological conditions.

**Author Contributions:** Conceptualization, J.D.-G.; methodology, J.J.P. and M.Á.L.-G.; software, J.C.P.-B.; formal analysis, J.D.-G. and J.J.P.; investigation, J.D.-G., J.J.P., M.Á.L.-G. and J.C.P.-B.; data curation, J.C.P.-B.; writing—original draft preparation, J.D.-G.; writing—review and editing, J.J.P.,

M.Á.L.-G., and J.C.P.-B.; supervision, J.J.P. All authors have read and agreed to the published version of the manuscript.

**Funding:** This work was supported by the Assistance to Research Groups (GR18102) of the Junta de Extremadura (Ministry of Employment and Infrastructure); through the European Regional Development Funds (ERDF). This research was supported by an FPU predoctoral grant from the Government of Spain (Ministry of Education, Culture and Sports) for J.D.G. (FPU18/03660). This research also received support from the Government of Spain (Ministry of Science and Innovation) for J.J.P. (IJC2019-040788).

**Data Availability Statement:** Not applicable.

**Acknowledgments:** 

**Conflicts of Interest:** The authors declare no conflict of interest.

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
