# Peer review of "Is Motivation Associated with Mental Fatigue during Padel Trainings? A Pilot Study"

_sustainability, doi:10.3390/su13105755_

Round 1

Reviewer 1 Report

After a thorough analysis of the article, I consider it suitable. 

Author Response

Thank for your review. We strongly think that your work has enhanced the quality of our manuscript. We have included all the information in the Word file.

Reviewer 2 Report

This is a very interesting article about mental effect related to padel training. Padel is becoming increasingly popular so I think this topic will become hot in the next future literature, so it is well welcome. 

I have only minor suggestions for you:

  • pay attention to english grammar and phrases all over the article
  • in the introduction section, I suggest to follow this structure: topic sentence > what is known in literature > what isn't known in literature > literature gap > scope of the study
  • please provide some limitations in your discussion section
  • I suggest you the following article for your reference section (https://www.mdpi.com/2411-5142/5/4/89) --> it is about training of reaction time and it proposes video observation for improving skills of  individuals who need to simultaneously develop a fast response to different types of stimuli: could it be useful for padel too? I think that a section upon proposed training programs for this new-born sport could be interesting for readers!

Author Response

(The authors gave the same response as above.)

Reviewer 3 Report

This research problem seems to be interesting and important.

The literature review is well-done and the research problem well-stated. The authors should also provide a description of the motivation theory they applied to this study.

The secondary objective is not clear. I would advise rephrasing it.

In Statistical Analysis, the tables perhaps should be clearer to make them more readable.

Some terms are not consequently used. Is the free lesson a reward or a constraint?

The authors should also perform an in-depth language revision.

The authors should also consider whether it is better to change "covariate" to "factor" to avoid problems with statistical terminology.

Since analyses on the distribution of variables were presented in the statistical analyses section,  it would be useful to refer to them in the result section as well.

Author Response

(The authors gave the same response as above.)
